# Surviving the Immediate Aftermath of a Disaster: A Preliminary Investigation of Adolescents’ Acute Stress Reactions and Mental Health Needs after the 2023 Turkey Earthquakes

**DOI:** 10.3390/children10091485

**Published:** 2023-08-31

**Authors:** Gökçe Yağmur Efendi, Rahime Duygu Temeltürk, Işık Batuhan Çakmak, Mustafa Dinçer

**Affiliations:** 1Department of Child and Adolescent Psychiatry, Şanlıurfa Mehmet Akif İnan Training and Research Hospital, Şanlıurfa 63500, Türkiye; dincermustafamd@gmail.com; 2Department of Child and Adolescent Psychiatry, Ankara University, Ankara 06590, Türkiye; rduygukaydok@gmail.com; 3Department of Psychiatry, Sungurlu State Hospital, Çorum 19300, Türkiye; batuhancakmak@hotmail.com

**Keywords:** acute stress disorder (ASD), earthquake, psychological needs

## Abstract

On 6 February, southeastern Turkey and parts of Syria were struck by two powerful earthquakes, one measuring a magnitude of 7.8 and the other, nine hours later, at a magnitude of 7.5. These earthquakes have been recorded as some of the deadliest natural disasters worldwide since the 2010 Haiti earthquake, impacting around 14 million people in Turkey. For trauma survivors, the stressors associated with an event can lead to the development of acute stress disorder (ASD) or other psychiatric disorders. Trauma experiences during adolescence can impact development and affect adolescents differently than adults. Although ASD in adults has been addressed in several studies, there is much less information available about how younger populations respond to acute stress. The aim of our study was to assess the occurrence of ASD among individuals seeking help at the Şanlıurfa Mehmet Akif İnan Research and Training Hospital Child and Adolescent Outpatient Clinic following the 2023 Turkey Earthquakes and the factors associated with acute stress reactions. A child and adolescent psychiatry specialist conducted psychiatric interviews with the adolescents, and the individuals were also asked to complete ‘The National Stressful Events Survey Acute Stress Disorder Short Scale’ (NSESSS) to evaluate acute stress symptoms. ASD diagnoses were established according to the Diagnostic and Statistical Manual of Mental Disorders, Fifth Edition (DSM-5) criteria. Results showed that 81.6% of the participants (*n* = 49) were diagnosed with ASD, and drug treatment was initiated in 61.7% of the cases (*n* = 37). It was determined that ASD rates did not differ according to gender, and patients without physical injury had higher acute stress symptom scores (*p* > 0.05). According to the logistic regression models, paternal educational levels and adolescents’ own requests for psychiatric assistance were predictors of acute stress disorder (OR 10.1, β = 2.31, *p* = 0.006 and OR 16.9, 95 β = 2.83, *p* = 0.001, respectively). Our findings revealed striking results in demonstrating the need for careful evaluation of adolescents without physical injury in terms of acute stress disorder and the need to pay close attention to the psychiatric complaints of adolescents willing to seek mental health assistance. Moreover, our study suggests that the proportion of adolescents experiencing acute stress symptoms after earthquakes might be higher than previously reported. Estimation of the incidence rate and symptoms of psychiatric distress in the short-term period following a disaster is important for establishing disaster epidemiology and implementing efficient relief efforts in the early stages. The outcomes of this study have the potential to yield novel insights into the realms of disaster mental health and emergency response policies, as well as their pragmatic implementations.

## 1. Introduction

In the early hours of 6 February, a powerful 7.8-magnitude earthquake struck southeastern Turkey and some parts of Syria. This was then followed by another 7.5-magnitude quake nine hours later, accompanied by over 200 aftershocks [1]. The first Pazarcık-centered earthquake was felt across a vast geography, including Turkey and Syria, as well as Lebanon, Cyprus, Iraq, Israel, Jordan, Iran, and Egypt. The two major earthquakes caused damage to an area of approximately 350,000 km^2^, affecting approximately 14 million people in Turkey, who constituted 16% of Turkey’s population [2]. These earthquakes, which occurred on the Eastern Anatolian Fault Line, caused a devastating impact, with over 50,000 people losing their lives in 11 provinces [3]. These earthquakes have also been recorded as some of the deadliest worldwide since the 2010 Haiti earthquake [4].

In the past two decades, natural disasters have resulted in millions of deaths worldwide, and hundreds of millions of people have suffered from various traumas [5]. These global disasters are characterized by their unpredictability and severity, encompassing a range of events, like earthquakes, nuclear meltdowns, pandemics, and food crises. Such catastrophic events frequently yield abrupt and overwhelming physical or psychological harm to populations, which can incite significant emotional distress and psychopathology [6]. For trauma survivors, the stressors associated with an event can lead to the development of acute stress disorder (ASD) and posttraumatic stress disorder (PTSD) or a number of other psychiatric disorders [7]. Researchers have extensively studied the psychological aftermath of natural disasters. Various studies have suggested that the percentage of individuals with psychiatric disorders could be as high as 60%, and PTSD rates might reach up to 74% following such events. However, findings have varied significantly between studies [8,9]. 

Childhood and adolescence are recognized as vulnerable periods for postdisaster psychological morbidity. For adolescents, experiencing trauma is especially significant, given the substantial physical and emotional growth during this phase. The stressors that adolescent encounters help shape their development and perspective and can have long-lasting impacts [10]. Traumatic events can influence the nervous and endocrine systems, even leading to structural changes in the brain due to severe stress [11]. Adolescence is also a time of social and emotional development. Trauma during this stage can result in social isolation, poor academic performance, and behavioral issues, all of which can impact both the current and future quality of life [12].

When a child or adolescent experiences a traumatic event, they may exhibit a variety of emotional responses [13,14]. While acute stress reactions in adults have been extensively studied, the understanding of how younger individuals respond to acute stress remains limited, as research has primarily concentrated on adults [15,16]. A recent meta-analysis examining acute stress disorder in children and adolescents evaluating 17 different studies reported that the evidence base is still quite limited and highly heterogeneous [17]. Recently, there has been a rise in studies focusing on acute stress symptoms among children and adolescents [18]. However, only a limited number of published investigations have explored acute stress symptoms in children and adolescents in relation to earthquake injuries [19]. 

ASD was introduced into the DSM-IV to describe acute stress reactions (ASRs) occurring in the initial month after exposure to a traumatic event and before the possibility of diagnosing PTSD and to identify trauma survivors in the acute phase who are at high risk for PTSD [20]. The diagnostic criteria underwent multiple changes with the introduction of the DSM-5 in 2013, and ASD was moved from the anxiety disorders category to a newly created one (i.e., trauma- and stressor-related disorders) to distinguish its characteristics further. Additionally, in contrast to DSM-IV, the diagnosis of ASD in DSM-5 no longer necessitates the presence of dissociative symptoms. While the accuracy of predicting subsequent PTSD cases is moderate, diagnosing ASD would be beneficial in acute trauma situations to identify individuals who may benefit from early interventions or ongoing monitoring [21].

In the aftermath of a disaster, it is common for the focus to be on fulfilling the community’s material and physical needs. Unfortunately, this can cause the psychological needs of children to be overlooked. To ensure their wellbeing, it is crucial to incorporate child mental healthcare into public health interventions for emergencies and disasters. To address the needs of children and adolescents after a natural disaster, it is crucial to assess their mental wellbeing and psychiatric symptoms during the acute phase after a disaster. Understanding and responding to disasters is crucial, and conducting research might provide valuable insights. It is essential to establish disaster mental health systems and capabilities in advance to ensure a swift response during the initial stages of such events.

Estimation of the incidence rate and course of psychiatric distress in the short-term period following a disaster is important for establishing disaster epidemiology and implementing practical relief efforts in the early stages [22]. The aim of our study was to assess the psychological distress and the occurrence of acute stress disorder among individuals seeking help at the Şanlıurfa Mehmet Akif İnan Training and Research Hospital following the 2023 Turkey Earthquakes. Furthermore, we explored sociodemographic, clinical, and event-related factors that could potentially correlate with acute stress reactions.

The main objective of this study was to examine the acute stress symptoms in adolescents during the immediate postearthquake period following the devastating Turkey earthquakes in 2023. Additionally, building on prior research, we aimed to assess certain risk factors that might potentially make earthquake survivors more susceptible to developing ASD. These factors encompassed variables such as bodily injuries, the educational backgrounds of adolescents and their parents, and the gender of the patients. The outcomes of this study are expected to provide valuable insights for informing policy formulation, enhancing disaster readiness protocols, and advancing mental health intervention strategies within Turkey and across global contexts.

## 2. Materials and Methods

### 2.1. Data Collection

Following the earthquake on 6 February 2023, child and adolescent psychiatry clinics in hospitals across several impacted provinces faced disruptions in their services for varying durations. The hospital where this study took place was located in Şanlıurfa, one of the affected provinces. However, the hospital building remained undamaged, allowing the child and adolescent psychiatry clinic to open its doors to all patients seeking psychiatric assessment without requiring an appointment in the aftermath of the earthquake. 

During the immediate aftermath of the earthquake, the principal investigator, who had not personally experienced the earthquake and was outside the city at the time of the incident, assumed responsibility for child and adolescent psychiatry outpatient services upon returning to Şanlıurfa. In contrast, the remaining child and adolescent psychiatrists at the hospital were unable to provide outpatient services for two weeks due to their direct exposure to the traumatic event. They had to deal primarily with their own vital needs and survival. While other child and adolescent psychiatrists resumed their duties after two weeks, the principal investigator continued to conduct psychiatric interviews for adolescents seeking help at the outpatient clinic. Other team members focused on providing assistance in different areas to children and adolescents affected by the earthquake, such as providing consultations for inpatients being treated for their injuries and providing families with education about the psychological effects of disasters.

Our study included children and adolescents aged 11–17 years who sought treatment between 7 February and 7 March 2023 at the Child and Adolescent Psychiatry Outpatient Clinic of Şanlıurfa Mehmet Akif İnan Training and Research Hospital. The study focused on evaluating the symptoms of patients who sought help within a month after the earthquake, as the primary goal of our study was to assess the psychiatric symptoms in children during the acute period following the traumatic experience. The reason for evaluating the symptoms of adolescents aged 11–17 in our study was to assess patients experiencing similar developmental stages. 

A child and adolescent psychiatry specialist conducted psychiatric interviews with the patients, and diagnostic evaluations were based on the DSM-5 manual. Sociodemographic data forms evaluating the patients’ experiences during and after the earthquake (accommodation in a place other than home, loss of housing, etc.) were completed by the parents of the patients and placed in the patient files. Adolescents aged 11–17 were also asked to fill out the DSM-5 ‘Severity of Acute Stress Symptoms National Stressful Events Survey Acute Stress Disorder Short Scale—Child Age 11–17’ (NSESSS) form to measure the severity of acute stress symptoms of the patients and to be used in follow-ups. 

Ethics approval was secured from Harran University’s committee. Patient data were retrospectively collected following proper procedures. Since the Turkish Ministry of Health requested that an additional document called the ‘disaster notification form’ be filled out in addition to the applications of the earthquake victims, the patient files were kept in great detail. The treating physician reopened patient files to extract data for research purposes, and the data were used solely for the research, providing that results would be published in a way that did not allow patients’ identification. The participants and their families were contacted and informed about the study, and their permissions were obtained. The patient files of all adolescents aged 11–17 who applied to the Child and Adolescent Psychiatry Clinic within the specified period after the earthquake were analyzed. A convenience sampling method was utilized, as all suitable cases were included over a specific time frame. There were no exclusion criteria for our study except for missing information in the patient files. Out of the 72 patients aged 11–17 who sought treatment at our outpatient clinic in the first month after the earthquake, 12 were excluded due to incomplete information in their files and forms. Missing data from the 12 excluded files were not included in the analysis.

### 2.2. Measurements and Procedures

The sociodemographic questionnaire was created by experts in the field of medicine, specifically child and adolescent psychiatrists, and included questions about demographic information and the adolescents’ experience of the earthquake. Our clinic routinely employs a standard sociodemographic form to ensure comprehensive patient records. Questions about earthquake experience were added to this form after the earthquake. Demographic information obtained included data such as the age and gender of the children, whether the children continued to formal education, the educational status of the child, the education and employment status of the parents, and whether the parents were divorced. The questionnaire asked adolescents about their experiences during and after the earthquake, including whether they had lost a family member or friend, suffered injuries, or had their house destroyed. The sociodemographic form also included questions about the mental health services that adolescents required following the earthquake. 

A brief ASD screening tool, the DSM-5 ‘Severity of Acute Stress Symptoms National Stressful Events Survey Acute Stress Disorder Short Scale—Child Age 11–17’ (NSESSS), was used to screen for the presence of ASD symptoms and their severity. The NSESSS is designed to be used in the initial evaluation and treatment of children and adolescents diagnosed with acute stress disorder or individuals with acute stress disorder symptoms. This 7-item measure is designed to be completed by the child, and each item of the scale asks the child to rate the severity of their acute stress disorder during the past seven days. The total score can range from 0 to 28, with higher scores indicating greater severity of acute stress disorder. The average total score reduces the overall score to a 5-point scale, which allows the clinician to think of the severity of the child’s acute stress disorder in terms of none (0), mild (1), moderate (2), severe (3), or extreme (4) [23]. A Turkish validity and reliability study of the scale was conducted by Sapmaz et al. in 2017, and it was shown that the Turkish version of the scale can be used reliably and validly, both in clinical practice and research [24]. 

Our hospital’s Child and Adolescent Psychiatry Clinic started to provide service again on the fourth day after the earthquake. Service was provided to all patients who applied during the first month after the earthquake, including those without an appointment. First, a child and adolescent psychiatrist filled out the patients’ sociodemographic forms by directing questions to the patients, and this process took approximately 10 min. Subsequently, adolescents were given an NSESSS form to fill out, which took about 5 min. Following this, the same doctor carried out psychiatric evaluations of the patients. The duration of these interviews varied but generally spanned between 40 min and an hour. On average, the evaluation of a single patient took about an hour. The information obtained was stored in the patients’ electronic files on the hospital’s computer system. A retrospective review of the patients’ records was performed by the physician who also performed psychiatric evaluations of the patients.

### 2.3. Statistical Analysis

The sample size of the study was determined utilizing the G*Power 3.1 program [25]. A previous study examining the diagnosis and symptoms of acute stress disorder in children under 18 in Turkey after the 1999 Marmara earthquake was used as a reference. In this study, 74.5% of the children who experienced the earthquake were diagnosed with ASD [26]. With a Type-I error set at 0.05 and a targeted test power of 1 − ß = 0.80, the required sample size for statistical analysis was calculated as 54.

The Statistical Package for the Social Sciences (SPSS) version 23.0 program package was used for the statistical analysis of the data. Descriptive data were presented as numbers and percentages for categorical variables (e.g., sociodemographic features, psychiatric symptoms, ASD diagnosis) and means and standard deviations represented numerical data (e.g., age, scale item scores). The Shapiro–Wilk test was used to determine the normality of data distribution. NSESSS scale scores comparisons among groups (divided into groups such as according to the reasons for applying to the child psychiatry clinic and according to gender) were analyzed using the Mann–Whitney U test because of the non-normal data distribution. Chi-squared and Fisher’s exact tests were used for the categorical variables, and a binary logistic regression model was conducted. All tests were two-tailed with a significance threshold of 0.05.

## 3. Results

### 3.1. Sociodemographic and Clinical Characteristics of Cases

The sociodemographic and clinical characteristics of the children are presented in Table 1. Upon examination of 16 adolescents with a family history of psychiatric illness, it was found that eight had psychiatric disorders in their mothers, four in their fathers, and four in their siblings. The most common psychiatric diagnosis was depression among parents. In contrast, psychotic disorders were the most common diagnoses in siblings.

Before the earthquake, 20 adolescents (33.3%) had consulted the Child and Adolescent Psychiatry Outpatient Clinic for support, while 40 (66.7%) had never sought psychiatric assistance. Among the 20 adolescents who had applied to child psychiatry before, 20% (*n* = 4) had attention deficit hyperactivity disorder, 20% (*n* = 4) had depressive disorder, 15% (*n* = 3) had anxiety disorder, and 15% had (*n* = 3) substance use disorder. Additionally, 15 (*n* = 3) of the patients who had previously sought psychiatric help were not formally diagnosed. It was found that 14 of the patients who had previously sought psychiatric assistance were using psychiatric drugs. The most frequently used drug combination was the combination of a selective serotonin reuptake inhibitor and an antipsychotic.

### 3.2. Earthquake-Related Experiences of the Cases

The experiences of children and adolescents related to the earthquake are given in Table 2. Additionally, upon examining the cases who lost family members in the earthquake, it was found that three had lost one parent, six had lost both parents and siblings, two had lost grandparents, and five had lost cousins.

Out of the total number of cases, 45 (75%) reported that their most distressing experience related to the earthquake was during the event itself. The remaining 15 (25%) found the media coverage and the images they saw afterward more unsettling than the actual earthquake.

Psychiatric symptoms of the patients after the earthquake are shown in Figure 1.

When the cases were examined regarding whether they had physical damage related to the earthquake, 46 (76.7%) of the cases had no bodily damage, 10 (16.66%) had a bone fracture, 2 (3.33%) had limb amputation, and 2 (3.33%) was found to have lung contusion. 

### 3.3. ASD Diagnosis and Pharmacological Treatments

During evaluations at the Child and Adolescent Outpatient Clinic following the earthquake, 49 (81.6%) of the cases received an ASD diagnosis. Of the remaining 11 (18.3%) cases, all exhibited acute stress symptoms but did not meet the DSM-5 criteria for an ASD diagnosis. Among the 49 patients diagnosed with ASD, 38 (77.5%) had solely ASD without any other coexisting conditions. Additionally, nine patients (18.3%) were diagnosed with both ASD and anxiety disorder, while two patients (4%) had both ASD and depressive disorder. Following psychiatric evaluation, drug treatment was initiated in 37 out of 60 cases, constituting 61.7% of the sample. Among the commonly prescribed drug treatments, the combination of selective serotonin reuptake inhibitors (SSRIs) and atypical antipsychotics (AAs) (20%) and AAs alone (20%) were the most frequently initiated. These were followed by SSRI (6.6%) and mirtazapine (6.6%) treatment alone.

### 3.4. NSESSS Scale Scores and Their Relationships with Different Variables

Among the 7 items of the NSESSS, Item 1 evaluates ‘flashbacks’, the reliving of past stress as if it is recurring; Item 2 assesses intense emotional distress triggered by reminders of stress; Item 3 gauges detachment from self, body, surroundings, or memories; Item 4 examines avoidance of stress-associated thoughts, feelings, or sensations; Item 5 measures sustained hyperalertness and vigilance for danger; Item 6 rates heightened startle response due to sudden noises; and Item 7 considers extreme irritability that might result in yelling, fights, or destructive behavior.

The mean values of the scores given to the 7 items and the total scale scores for the NSESSS are shown in Table 3. According to these results, on average, the highest score was given to the sixth item on the scale, an item questioning being easily startled and frightened. The third item obtained the lowest score, and this item questioned whether the person experienced the feeling of being dissociated from their own body or environment. In addition, while 35 (58.4%) of the participants rated the sixth item as ‘quite a lot/extremely’, 15 (25%) gave the same answer for the third item. 

Participants were divided into various groups and evaluated in terms of NSESSS scores. Initially, they were separated based on their reasons for seeking help at the Child Psychiatry Clinic (self-requested counseling or family-initiated). Further analysis divided adolescents into two groups according to gender and physical injury. After assessing the normality of data distribution using the Shapiro-Wilk test, the Mann–Whitney U test was conducted due to the non-normal data distribution (*p* < 0.05). Afterward, analyses were performed by dividing the patients into two groups according to their mean NSESSS scores, as those who scored below two on the NSESSS scale (no acute stress symptoms and mild acute stress symptoms) and those who scored two and above (moderate, severe, and extreme acute stress symptoms). Gender differences between these two groups were examined categorically with the chi-squared test.

When the groups were examined according to the reasons for applying to the Child Psychiatry Clinic, the total score of the NSESSS was found to be significantly different between the groups (Mann–Whitney U test, Z = −3.29, U = 209, *p* < 0.001). Adolescents who requested psychiatric counseling on their own demand had a higher score on the scale ([Mdn (IQR) = 16 (13–19)], 95% CI = 14.24–17.26) than those who were requested to seek psychiatric counseling by their families (Mdn (IQR) = 9 (5–17), 95% CI = 7.57–13.03). No significant difference was found between the mean scores of the NSESSS between male and female patients (Mann–Whitney U test, Z = −0.38, U = 400, *p* = 0.69; Mdn (IQR) = 16 (11.5–20.5), 95% CI= 10.92–15.85 for male, Mdn (IQR) = 15 (10.5–19.5), 95% CI = 11.82–15.85 for female). We used the Mann–Whitney U test to investigate whether there were differences in the NSESSS scores between patients with and without bodily injury. Our findings revealed that patients without bodily injury had significantly higher NSESSS scores (Mann–Whitney U test, Z = −2.29, U = 191, *p* = 0.02); Mdn (IQR) = 16 (9.75–19), 95% CI = 12.86–16.35 for without, Mdn (IQR) = 11 (5–14.25), 95% CI = 7.77–13.37 for with bodily injury).

Further analyses were performed by dividing the patients into two groups according to their mean NSESSS scores, as those who scored below 2 on the NSESSS scale (no acute stress symptoms and mild acute stress symptoms) and those who scored 2 and above (moderate, severe, and extreme acute stress symptoms). A total of 27 of the adolescents (45%) had a mean NSESSS total score of less than 2, whereas 33 (55%) scored 2 or more. No statistically significant difference was observed according to gender in terms of receiving a score of less than two and a score of two or more according to the total mean score of the NSESSS (chi-squared test, *p* = 0.85).

### 3.5. Predictors of Acute Stress Disorder

A binary logistic regression was carried out to determine further sociodemographic and clinical factors that might be associated with acute stress disorder scores as evaluated by the NSESSS. After the univariate analyses, the reasons for applying to the child psychiatry clinic, paternal educational level, current use of any psychiatric medicine, and the status of attending formal education were included in the regression model (*p* values < 0.10). Multicollinearity was checked using the variance inflation factor (VIF), and no problems were identified (i.e., VIF < 10). The current model, including these predictors, was significant (*p* < 0.001 and Nagelkerke R^2^ = 0.47), and the Hosmer–Lemeshow goodness-of-fit test was insignificant [X^2^ (8)  =  11.051, *p*  =  0.087], suggesting that the model fit the data well. According to the logistic regression model presented in Table 4, paternal educational levels and reasons for applying to the Child Psychiatry Clinic were predictors of acute stress disorder.

## 4. Discussion

While there are various studies in the literature regarding the psychiatric disorders experienced by individuals after an earthquake, there is a scarcity of research focused on both the acute period following the earthquake and adolescents. Our study aimed to fill the gap in the existing literature by assessing the psychological symptoms and needs of adolescents who sought assistance at a child psychiatry clinic within the first month after experiencing an earthquake.

### 4.1. Psychiatric Symptoms of the Adolescents

In this study, we evaluated 60 earthquake-affected adolescents and identified fear/anxiety as the most prevalent psychological symptom. Among the patients, 41.6% (*n* = 25) experienced fear/anxiety, followed by equilibrium disturbances/dizziness at 31.6% (*n* = 19). In the literature, different studies dealing with postearthquake somatic and psychological symptoms have reported a remarkable increase in the patients presenting with vague, dizziness-like features, which cannot be attributed to any defined variant of vestibular disorder. Nomura et al. conducted an epidemical clinical study and labeled earthquake-associated dizziness as ‘post-earthquake dizziness syndrome’ following a major earthquake in Japan on 11 March 2011 [27]. In a later study conducted in Japan, 36.4% of participants reported experiencing postearthquake dizziness, and changes in living conditions and autonomic stress were found to be associated with dizziness symptoms [28]. In another study conducted with adolescents after an earthquake in China, the percentage of patients experiencing dizziness in the third month after the earthquake was found to be 41.9%, and this rate decreased to 34.8% in the sixth month. Postearthquake dizziness/equilibrium disturbance rates reported in our study are relatively lower than those reported in previous studies, and to gain a deeper understanding of this phenomenon and the progression of the symptoms, our team is closely monitoring the patients in this group through ongoing regular follow-ups.

To our knowledge, our study is one of the few studies reporting postearthquake vertigo symptoms in adolescents, and although this phenomenon has been studied and documented in adults, our research sheds new light on its occurrence in younger individuals. There is currently no established specific treatment approach for these symptoms; however, the appropriate management of such earthquake-induced psychological stress resulting in dizziness should encompass interdisciplinary assessment, appraisal of the underlying impairment, and appropriate counseling and therapeutic approaches [29]. 

### 4.2. The Prevalence of ASD 

In this study, ASD diagnosis was assessed through clinical interviews based on DSM-5 criteria conducted by a child and adolescent psychiatrist, and a significantly higher rate than the rates of ASD reported in the literature following earthquakes was detected. In contrast to earlier findings, we found that 81.6% (*n* = 49) of cases were diagnosed with ASD. A study conducted in India reported that 48% of children aged eight years and older who visited the emergency department of a hospital after an earthquake scored above the cut-off value for stress-induced psychological disorder diagnosis on the Children Impact of Event Scale [30]. Another study evaluating acute stress symptoms in children after the 1999 earthquake in Turkey with a semistructured clinical interview reported that 74.5% of children and adolescents were diagnosed with ASD [26]. 

Our study found a higher percentage of patients diagnosed with acute stress disorder compared to other studies. One potential explanation for this disparity could be attributed to the severity and destructiveness of the two major earthquakes, along with the subsequent aftershocks. Gökçen et al. showed that even a moderate-intensity earthquake without any devastation might cause significant PTSD symptoms in children and adolescents; however, previous studies have indicated that individuals exposed to higher levels of trauma are likely to develop posttraumatic stress reactions [31,32]. Research indicates that previous traumatic events can be a predictor of acute stress reactions following a current traumatic experience, and it may be suggested that exposure to two major earthquakes on the same day might have exacerbated acute stress symptoms in patients by creating an additive traumatic effect [33]. Moreover, the earthquakes received extensive coverage from both national and international media. Graphic images were also shared on traditional and social media about individuals who had family members trapped under debris or who had lost their lives. Considering the fact that 25% of the patients in our study stated that what they saw in the media after the earthquake was more traumatic than the moment of the earthquake, it can be argued that, especially, the graphic images shared on social media may partly explain the high rate of ASD in our study. There are studies reporting that the interaction of media exposure with emotional reaction to media coverage in the aftermath of traumatic events might predict ongoing posttraumatic stress, and research conducted after Hurricane Sandy revealed that individuals who used social media experienced higher stress levels than those who relied solely on traditional media [34,35]. Some findings suggested that higher social media exposure was associated with an increased likelihood of experiencing acute stress symptoms, such as anxiety, fear, and sleep disturbances [36]. Although media use habits and intensities of adolescents were not examined in our study, it seems reasonable to suggest that there may be a correlation between media use and posttraumatic stress symptoms. Further research is necessary to comprehend the impact of viewing trauma-related news or images in the media on stress symptoms.

### 4.3. The Initiation of Pharmacological Treatments

Drug treatment was initiated in 61.7% of the 60 patients evaluated in our study, and the most frequently used drug groups were determined as atypical antipsychotics with a combination of SSRIs and atypical antipsychotics alone. There is a limited amount of research on the use of medication to treat ASD, as most studies on children and adolescents focus on the treatment of PTSD. In various treatment guidelines dealing with ASD treatment, pharmacological treatments are not recommended for use as an early intervention for ASD or related conditions [37]. One meta-analysis of pharmacological interventions for the prevention of PTSD found that, across 14 studies, pharmacological interventions were effective in treating ASD or preventing PTSD; however, no effect was found when only randomized, controlled trials were included [38]. While existing pharmacological trials do not provide strong evidence for specific medications to treat ASD currently, there are studies in the literature that show some benefits when initiating pharmacological treatments for individuals with ASD. These benefits might vary depending on a patient’s condition. In certain cases, the use of medication is highlighted for managing acute symptoms effectively [39]. A randomized, double-blind clinical study using imipramine in pediatric burn patients suggested that imipramine may be cautiously used to reduce symptoms of ASD [40]. Another study showed that risperidone, an atypical antipsychotic, may be effective in relieving ASD symptoms in children with burns [41]. In addition, although large-scale psychopharmacology studies have not been conducted in the literature for the treatment of trauma-related disorders in children and adolescents, Cohen et al. reported that 95% of child psychiatrists had used pharmacotherapy to treat childhood and adolescent PTSD, and the medications most frequently used were selective serotonin reuptake inhibitors and α-adrenergic agonists [42]. To the best of our knowledge, our study is the first in the literature to report the rate of drug initiation and treatments in clinical practice for adolescents diagnosed with ASD. Large-scale and longitudinal further studies are needed to evaluate factors affecting clinicians’ drug choices and the efficacy of medication used off-label in children and adolescents with ASD.

There are few studies in the literature regarding the effectiveness of nonpharmacological interventions in managing the symptoms of acute stress disorder. Although psychological debriefing (PD), one of the nonpharmacological interventions for acute stress symptoms, has been frequently applied to adolescents and adults after traumatic events in the past, it was recently abandoned as it was revealed that it was not helpful [43]. There are more studies on the effectiveness of nonpsychopharmacological interventions for PTSD symptoms in children and adolescents. Various studies have shown that cognitive behavioral therapy (CBT) and eye movement desensitization and reprocessing (EMDR) treatments can be particularly effective in the treatment of PTSD symptoms in children and adolescents [44,45]. Different global mental health organizations recommend the use of psychotherapy in the treatment of PTSD. For instance, the American Academy of Child and Adolescent Psychiatry (AACAP) recommends that trauma-focused psychotherapies should be used as the primary frontline treatment for PTSD in children and adolescents [46]. The most recent edition of the Australian guidelines for preventing and treating PTSD emphasizes the use of trauma-focused cognitive behavior therapy for children with PTSD, either alone or with a caregiver [47]. 

We are currently maintaining contact with the patients who participated in our study, continuing their care at our clinic. Some patients have been referred to psychologists for CBT treatment, as per recommended guidelines. Notably, the referral of the patients to psychologists commenced a month after the earthquake. This delay was due to hospital psychologists facing personal losses and being unable to return to work immediately during the initial weeks postearthquake. As our study concentrated on evaluating acute psychiatric symptoms following the earthquake, assessments of nonpharmacological treatments were not included. Findings on nonpharmacological interventions will be presented in our upcoming follow-up study.

### 4.4. Evaluation of the Scores in the Acute Stress Scale

In our study, we evaluated the responses to the NSESSS and found that the sixth item, an item asking about feeling jumpy or being easily startled upon hearing unexpected noises, had the highest average score, and 58.4% of the participants rated experiencing this sixth item as ‘quite a lot/extremely’. A previous study using data from 15 studies assessing acute stress symptoms conducted with children and adolescents from 5 to 17 years of age reported that 36.3% of the participants experienced hypervigilance symptoms and about 24.7% experienced exaggerated startle responses [48]. Studies evaluating the posttraumatic stress symptoms of adolescents in the literature have mainly focused on PTSD, and symptoms regarding being easily startled and frightened have been reported frequently. A previous study examining adolescents’ postdisaster experiences and psychiatric problems 13 months after the 1999 Marmara earthquake in Turkey showed that 68% of the participants experienced startling easily [49]. In addition, a study describing the posttraumatic stress disorder symptoms in adolescent survivors three months after Wenchuan Earthquake reported that 49.1% of the participants experienced symptoms of increased arousal [50]. Although studies evaluating acute stress symptoms in adolescents have reported hypervigilance symptoms at different rates, it is evident that many adolescents experience feeling jumpy or being easily startled after a traumatic experience. Zhang et al. showed that the arousal symptom cluster was one of the most influential predictors of future PTSD among trauma-exposed children and adolescents; therefore, recognizing and following up on children experiencing this cluster of symptoms might be particularly important [51]. Adolescents evaluated within the scope of our study will also be followed closely, and the relationship between this symptom cluster and PTSD will be further investigated and reported.

It was determined that the third item was the subscale that the participants in our study gave the lowest average score in the NSESSS, and only 25% of the participants rated the third item, an item questioning feelings of being detached or distant from oneself or one’s physical surroundings or memories, as ‘quite a lot/extremely’. In accordance with the present results of our study, previous research demonstrated that the requirement of dissociative symptoms for a diagnosis of ASD in the DSM-IV was stringent, and most patients failed to meet complete DSM-IV ASD diagnosis criteria because of lacking dissociative symptoms [52]. Furthermore, in another study, acute stress disorder minus dissociation symptoms was reported to be almost three times more sensitive in adolescents than complete DSM-IV acute stress disorder criteria in predicting later PTSD [53]. The criteria for ASD in the DSM-5 have been modified due to evidence showing that posttraumatic reactions vary greatly and that the DSM-IV’s focus on dissociative symptoms was too narrow [54]. 

One of the remarkable results of our study is that the NSESSS scores of the adolescents who applied to the Child Psychiatry Outpatient Clinic at their own request were significantly higher than those who were brought for counseling at the request of their families. In addition, the logistic regression analyses performed determined that an adolescents’ demand to apply to the psychiatry clinic predicted a clinical diagnosis of ASD. This discovery holds significant importance, as it highlights the necessity of recognizing adolescents’ expressed need for psychiatric assistance and is consistent with some previously reported results from other studies. A previous study conducted to examine and identify predictors of ASD and ASD symptomatology in children hospitalized for injuries reported that parents might underestimate their children’s acute distress [55]. Another study conducted by Meiser-Stedman et al. examining parent–child agreement for ASD and PTSD in children and adolescents reported that parent–child agreement for ASD was poor and child-reported ASD predicted later child-reported PTSD, while parent-reported ASD failed to predict later parent-reported PTSD. Furthermore, the same study determined that parent-reported ASD failed to predict later child-reported PTSD, suggesting that parent-only screening in the aftermath of trauma would be less than optimal [56]. In addition, a study conducted with children between ages 8 and 17 hospitalized for their injuries showed that the parent–child agreement was low for ASD diagnosis in children, and parents without ASD underestimated their child’s ASD compared to the child’s self-rating [57]. Considering both this study and previous studies, it can be posited that, irrespective of parental perceptions regarding the impact of traumatic events on their children, attending to adolescents’ articulated desire for psychiatric support assumes paramount importance. Our study, which reported findings supporting previous research, contributes to the literature in terms of emphasizing the importance of listening to adolescents who express their need for psychiatric help.

### 4.5. Physical Injury and Acute Stress

A noteworthy finding from our study is that the NSESSS scores of adolescents who did not suffer physical injuries were notably higher than those who did. Higher mean scores on the NSESSS indicate a more intense experience of acute stress symptoms. Although many studies in the literature have shown a significant relationship between the severity of PTSD symptoms and the presence of physical injury, studies addressing the relationship between ASD and bodily injury in adolescents are scarce [58,59,60]. A study conducted in Turkey after the Marmara earthquakes showed that ASD and PTSD were more common in children who sustained bodily injuries, and this result was subsequently corroborated by a limited number of investigations examining the association between physical injuries and ASD symptoms in adolescents [61,62]. One possible explanation for children without physical injury having higher NSESSS scores than those with bodily injuries in our study might be that children with injuries were in contact with hospital staff while receiving medical care. This communication might have provided children with a better chance to process their trauma. Haag et al. reported that having been treated as an inpatient predicted less severe ASD in children with road traffic accidents, and injury severity was not found to be predictive of ASD severity. To explain their findings, which are similar to our results, the researchers proposed the possibility that children admitted to the hospital received more professional care, and the process of inpatient treatment helped ease the cognitive and emotional processing of the traumatic event [63]. In addition, children hospitalized for a short time due to their physical injuries were temporarily away from the images and news related to the earthquake in the external environment and the media, and it can be argued that this may be associated with lower NSESSS scores. Although more comprehensive adolescent studies addressing the relationship between physical injury and acute stress symptoms are needed, the findings of our research are important for a number of reasons. After massive natural disasters such as earthquakes, resources often focus on people with physical injuries; however, as our findings indicate, it is crucial to evaluate the psychological wellbeing of adolescents, even if they have not sustained physical injuries. Additional programs that carry child and adolescent mental health services outside of hospitals may be beneficial to reach young people after an earthquake who do not have physical injuries. After the 2023 earthquakes, mobile ‘psychosocial support’ tents were set up by the Ministry of Health of the Republic of Turkey in settlements in many provinces affected by the event, and our findings emphasize the importance of such interventions. Moreover, providing trauma-sensitive training to healthcare professionals can bring about added advantages, as the way children who are hospitalized for physical injuries interact with a treatment team may help alleviate any posttraumatic psychological symptoms they may be experiencing.

### 4.6. Educational Status of Fathers and Acute Stress Disorder in Adolescents

Another thought-provoking result of our study is that we showed that the fathers’ education level predicted the diagnosis of ASD in adolescents. Although there are various studies in the literature reporting a relationship between fathers’ education levels and PTSD symptoms in children and adolescents, to the best of our knowledge, no studies in the literature have shown a correlation between a father’s education level and ASD symptoms. El-Khodary et al. reported that lower paternal education predicted a PTSD diagnosis in Palestinian children and adolescents, and they argued that parents with lower educational levels might have lower family income and more economic pressure [64]. Similarly, in a study conducted in Greenland, the low education level of the father was found to be a significant predictor of PTSD in adolescents [65]. Research has also demonstrated that, when fathers have lower levels of education, their children have a higher likelihood of experiencing various traumas [66,67]. However, to the best of our knowledge, our study is the first study in the literature to report that an increase in a father’s education level predicts ASD. One factor that might explain this noteworthy result may be the relationship between the level of education and the involvement in interaction with children and the family-related processes of fathers. Marsiglio found that fathers with higher education tended to report higher levels of paternal involvement with their school-aged children, and other studies have also reported that older and more educated fathers tend to be more highly involved with their children [68,69,70]. Although father involvement generally affects children’s mental health and development positively, it has been shown that, in some cases, it can also have adverse effects. Liu et al. reported that, at high levels of paternal inconsistency, higher father involvement was associated with higher behavioral and emotional problems in children, and when father–child relationships were poor, higher father involvement was also related to more behavioral and emotional problems [71]. Based on these previous findings, it can be suggested that fathers with higher education in our study were more involved in family relations and childcare processes. Our study did not examine the earthquake’s psychological impact on fathers. However, if fathers were affected and displayed signs of acute stress, it could have impacted the acute stress symptoms of their children. This may have been more pronounced in children whose fathers had a higher education level, as they may have had more interaction with their fathers. Studies have shown that there is a link between lower levels of education among fathers and trauma-related disorders, such as PTSD, in children and adolescents [72,73]. Our study emphasizes the significance of recognizing that children of highly educated fathers may also be at risk of acute stress disorder and that they should receive thorough evaluations and attention. It would be beneficial to conduct further studies that explore the factors associated with the link between fathers’ education levels and ASD in adolescents. Such studies would help shed more light on this phenomenon and enhance our understanding.

### 4.7. Limitations

The implications of this study’s findings should be considered within the context of certain limitations. Our assessment was restricted to adolescents who sought care at the outpatient clinic, thereby implying that caution should be exercised when attempting to generalize the findings to the entire population. Large-scale future studies on this subject should focus on a population sample rather than a clinical sample to achieve results that can apply to the general population. 

As mentioned previously, the absence of an examination of ASD symptoms in the parents could introduce an additional limitation to the study. Considering the severity and spread of the natural disaster experienced, it can be predicted that at least some parents may have psychiatric symptoms related to posttraumatic stress. Research has demonstrated that the stress symptoms and psychological resilience of parents can impact their children’s posttraumatic psychiatric symptoms [74,75]. To better understand and address acute stress symptoms in adolescents following natural disasters, it would be helpful to examine the psychological reactions of parents to trauma and how these reactions may impact their children’s psychiatric symptoms.

In addition, one of the study’s limitations is that information about past traumas, which may affect patients’ current trauma responses, was not comprehensively addressed in the patient evaluation process. Studies indicate that previous traumatic experiences may predict PTSD after subsequent traumatic events [76]. To gain a deeper understanding of how adolescents react to stress and trauma, it would be useful to conduct future studies that take into account previous traumatic experiences of adolescents. This could provide valuable insight into the topic.

Lastly, the cross-sectional design only captured data at a specific point in time, providing a snapshot of the participants’ experiences without accounting for potential changes in ASD symptoms over time. Longitudinal studies would be needed to better understand the trajectory and persistence of ASD symptoms in this population following earthquake exposure. We will conduct a longitudinal evaluation of the patients in our study to understand better the long-term course of ASD symptoms in adolescents and associated factors, and we plan to report the results in a future research publication. Despite these limitations, the study provides valuable insights into the immediate effects of earthquakes on adolescent mental health, highlighting the need for further research in this area to inform targeted interventions and support strategies for this vulnerable population.

## 5. Conclusions

In conclusion, the current study contributes to the existing literature by addressing the psychological symptoms of adolescents seeking assistance at a child psychiatry clinic shortly after experiencing an earthquake. Our research adds to the existing body of knowledge by highlighting the importance of assessing teenagers for acute stress disorder who have fathers with high levels of education and no history of physical injury. This study also provides valuable insights into the psychological impact of earthquakes on adolescents and emphasizes the need for targeted interventions, such as psychosocial training of hospital staff about trauma-informed care and the utilization of mobile psychosocial support tools, in this vulnerable population. As the world is facing increasing environmental disasters, developing effective postdisaster trauma response programs and establishing trauma-related policies are critical, not only for people in developing countries but also for people worldwide. Comprehending acute stress disorder in adolescents, who constitute a significant proportion of the world population, following natural disasters is of utmost importance, yet the information on this subject is currently scarce. Longitudinal studies that examine patients from various sociodemographic and cultural backgrounds are necessary to understand this subject comprehensively, as well as treatment and follow-up studies. We believe that this study, as well as further research built upon our findings, can guide the development of comprehensive strategies and mental health policies to address the mental health needs of adolescents affected by earthquakes and other traumatic events.

## Figures and Tables

**Figure 1 children-10-01485-f001:**
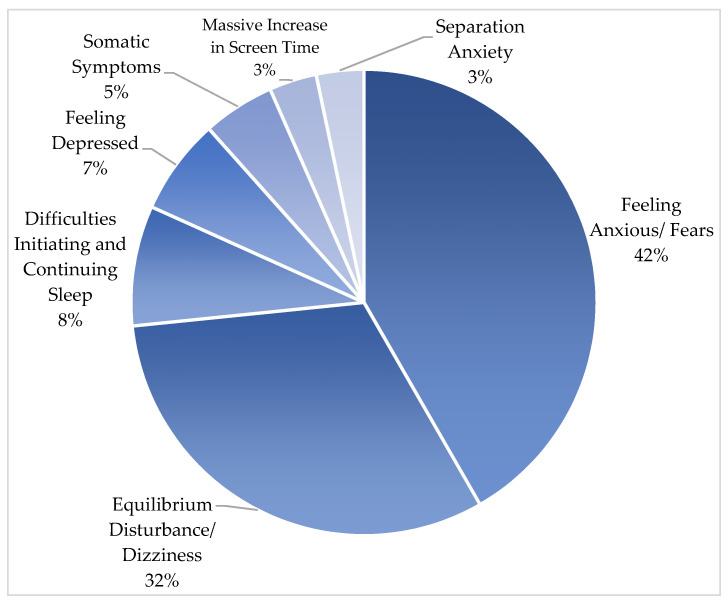
Psychiatric symptoms of the cases.

**Table 1 children-10-01485-t001:** Sociodemographic and clinical characteristics of cases.

Sociodemographic Variables	Participants (*n* = 60) Mean ± SD (Min–Max)/*n* (%)
Gender	
Female	37 (61.7)
Male	23 (38.3)
Age (years)	13.73 ± 1.86 (11–17)
Attending Formal Education	
Yes	50 (83.3)
No	10 (16.7)
Maternal Education, *n* (%)	
Literate	17 (28.33)
Primary school	24 (40)
Secondary school	5 (8.33)
High school	10 (16.66)
College degree or higher	4 (6.66)
Paternal Education, *n* (%)	
Literate	2 (3.33)
Primary school	31 (51.66)
Secondary school	4 (6.66)
High school	16 (26.66)
College degree or higher	7 (11.66)
Maternal Occupation, *n* (%)	
Housewife	57 (95)
Laborer	2 (3.3)
Civil servant	1 (1.7)
Paternal Occupation, *n* (%)	
Unemployed	6 (10)
Laborer	27 (45)
Civil servant	8 (13.3)
Tradesman	14 (23.3)
Farmer	5 (8.3)
Family Type, *n* (%)	
Parents married and living together	54 (90)
Single-parent family	6 (10)
Number of Siblings	2.81 ± 1.45 (0–6)
Presence of Individuals with Psychopathology in the Family	
Absent	44 (73.3)
Present	16 (26.7)

SD: Standard deviation; Min: minimum; Max: maximum.

**Table 2 children-10-01485-t002:** Earthquake-related experiences of the cases.

Earthquake-Related Experiences	Participants (*n* = 60) *n* (%)
City of Residence	
Şanlıurfa	43 (71.7)
Other cities	17 (28.3)
Migration to Another City After the Earthquake	
Yes	20 (33.3)
No	40 (66.7)
Accommodation Outside the Home After the Earthquake (Ever)	
Yes	54 (90)
No	6 (10)
Accommodation Outside the Home After the Earthquake (Still at the Time of Evaluation)	
Yes	22 (36.7)
No	38 (63.3)
Damage Status of the House	
No damage	20 (33.3)
Slightly damaged	22 (36.7)
Medium–heavy damaged	3 (5)
Collapsed in the earthquake	15 (25)
Loss of Family Members in the Earthquake	
No	44 (73.3)
Yes	16 (26.7)
First-degree relatives	9 (15)
Others	7 (11.7)

**Table 3 children-10-01485-t003:** The mean values of the scores given to the items in the NSESSS.

Scale Items	Participant (*n* = 60) Mean ± SD
Total scale score	1.90 ± 0.83
Item 1	2.11± 1.15
Item 2	2.30 ± 1.26
Item 3	1.41 ± 1.25
Item 4	1.65 ± 1.31
Item 5	2.16 ± 1.35
Item 6	2.53 ± 1.29
Item 7	1.48 ± 1.08

SD: Standard deviation.

**Table 4 children-10-01485-t004:** Logistic regression model for prediction of acute stress disorder.

	β	SE	*p*	OR (95% CI)
Reason for applying to the Child Psychiatry Clinic	2.83	0.88	0.001	16.94 (2.99–96.01)
Paternal educational level	2.31	0.84	0.006	10.16 (1.94–53.27)
Current use of any psychiatric medicine	1.24	0.84	0.143	3.45 (0.65–18.18)
Status of attending formal education	−1.77	0.95	0.065	0.17 (0.02–1.11)

SE: standard error; OR: odds ratio.

## Data Availability

Readers can access the data used in this study upon request from the corresponding author.

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
