# Peer review of "Surviving the Immediate Aftermath of a Disaster: A Preliminary Investigation of Adolescents’ Acute Stress Reactions and Mental Health Needs after the 2023 Turkey Earthquakes"

_children, 2023, doi:10.3390/children10091485_

Round 1

Reviewer 1 Report

1. In the introduction, it is recommended that the author include further details regarding the theoretical construction and development immediately following a disaster, as well as the acute stress reactions and mental health needs of adolescents.

2. Within the measurement section, additional information concerning the theoretical framework should be provided, including a literature review of the tools or items employed in this study.

3. Enhancing the statistical analysis within the results section is advised to ensure more robust significance testing, utilizing methods like t-tests, ANOVA, or correlation analysis.

4. Consider relocating certain discussions regarding theoretical literature from the discussions section to the introduction for better coherence.

Reviewer 2 Report

Dear Authors

Thank you for submitting your manuscript, “Surviving The Immediate Aftermath of a Disaster: A Preliminary Investigation of Adolescents' Acute Stress Reactions and Mental Health Needs After the 2023 Turkey Earthquakes." Your work contributes valuable insights to the field and provides a much-needed perspective on this crucial issue.

Upon careful review, we have identified some areas in the paper that could benefit from revision to enhance clarity, comprehensiveness, and impact. Here are our recommendations:

Abstract:

  • Background Context: More context could be provided about the significance of studying ASD in adolescents, especially when compared to adults. What makes this population particularly important or unique?
  • Methodological Details: The abstract could benefit from a brief mention of the methodology used to diagnose ASD or to initiate drug treatments. What criteria or methods were applied?
  • Clarity in Statistical Findings: The statistical findings might be presented more clearly, briefly explaining what the numbers mean for readers who may need to become more familiar with the specific terminology or concepts.
  • Key Insights and Implications: A brief summarization of the most significant findings and their implications for disaster response, mental health practice, or policy could provide a more robust conclusion to the abstract.
  • Language and Flow: Consider revising the language to improve flow and readability. Some sentences are quite dense, and breaking them down or rephrasing them may enhance comprehension.
  • Specificity in Recommendations: The final statement about the importance of estimating the incidence rate of psychiatric distress could be tied more directly to the study’s specific findings. How do the study's results specifically inform disaster epidemiology or relief efforts?
  • Abbreviation Explanation: If "ASD" is used for the first time in the abstract, it might be helpful to spell out "acute stress disorder" alongside the abbreviation to ensure all readers understand what it refers to.

The Introduction section provided here is detailed and does an excellent job in setting the context for the research study, including the description of the earthquake and the relevant background information related to acute stress disorder (ASD) and post-disaster mental health. However, there are a few areas that could be enhanced to make this section more impactful and precise:

  • Improve Cohesiveness and Flow: The information jumps from one topic to another. Organising the content into well-defined paragraphs with smooth transitions would improve readability. For example, one section might cover the specifics of the earthquake and the immediate impact, while another might detail the general effects of disasters on mental health, focusing on ASD.
  • Focus on the Target Population: Since the study concentrates on adolescents, highlighting this population's unique vulnerability or needs early in the introduction could create a stronger connection to the research question.
  • Clarify the Novelty or Gap in Research: It's mentioned that there is less information available about how younger populations respond to acute stress. Expanding on this and articulating the existing gap in more detail would provide more substantial justification for the study.
  • Explicit Statement of Objectives: While the aim is mentioned at the end of the introduction, a more formal statement of the research objectives, hypotheses, or research questions might make the focus of the study more straightforward. This might include detailing the specific aspects of ASD that will be explored or the assessment methods.
  • Improve Language and Conciseness: Some sentences are pretty long and complex, making them difficult to follow. Breaking these into shorter sentences and using more concise language might enhance understanding.
  • Emphasize the Importance of the Study: Including a statement that emphasizes the practical importance of the research (e.g., for policy-making, disaster preparedness, and mental health interventions) can highlight the study's relevance and potential impact.

The Materials and Methods section you provided is quite thorough in explaining the process of the study. However, there are still some aspects that can be improved or clarified to enhance the understanding and reproducibility of the research:

  • Sampling Methodology: There should be a clear explanation of the sampling methodology. Was this a convenience sample, or were specific criteria used to select the patients? Including details about the sampling method can clarify potential biases and the generalizability of the findings.
  • Inclusion and Exclusion Criteria: Though it's mentioned that 12 were excluded due to incomplete information, providing detailed inclusion and exclusion criteria would add more transparency to the study design.
  • Ethical Considerations: While the ethics committee approval is mentioned, further details on the consent process for both the patients and their parents might enhance the ethical transparency of the study.
  • Detailed Description of Tools: The section mentions the tools used (such as the sociodemographic questionnaire and the NSESSS scale), but it could provide more details about why these particular tools were chosen and how they were administered.
  • Inter-rater Reliability: If interviews and evaluations were conducted, information about the inter-rater reliability (if multiple specialists were involved) would be useful to gauge the consistency of the assessments.
  • Procedure and Timing: A brief step-by-step rundown of the procedures, including timing (such as the duration of interviews or how quickly after the earthquake these assessments were made) would provide additional context.
  • Potential Biases and Limitations: Acknowledging potential biases in the data collection process, such as observer biases or response biases from the participants, would provide a more balanced view of the methodology.
  • Data Handling and Confidentiality: Explaining how data was stored, handled, and kept confidential would add to the ethical integrity of the study.
  • Sample Size Justification: Adding a section that justifies the chosen sample size (including power analysis or other rationale) would enhance the understanding of how the sample size relates to the study goals.
  • Missing Data Treatment: It would be useful to include information about how missing data (such as the 12 excluded files) was handled in the analysis.
  • Statistical Models Details: While the statistical tests are outlined, adding more details about why specific tests were chosen, the underlying assumptions of those tests, and how they align with the research questions would strengthen this section.
  • Multicollinearity Check: If a binary logistic regression model were conducted, mentioning whether a multicollinearity check was performed would be beneficial to understanding the model’s validity.

Results: what could be improved:

  • Clarity and Organization:
    • Breaking down the large text into subsections with clear headings would help readers navigate the results.
    • A brief summary or interpretation of the results could make it more accessible for a broader audience.
    • It would strengthen the research by providing more details on the statistical methods used, such as what specific tests were applied, why they were chosen, and how they were interpreted.
    • Including confidence intervals and p-values in the results would offer a more comprehensive view of the significance.
  • Language and Style:
    • Some sentences are pretty long and complex. Breaking them down into shorter, more straightforward sentences might improve readability.
    • Ensuring consistent terminology and avoiding jargon or overly technical language can make the text more accessible.
  • Discussion:

    Here are some potential areas for improvement:

    • Structure and Clarity:
      • The text might benefit from subheadings that delineate various aspects, like "Prevalence of Anxiety," "Post-Earthquake Dizziness Syndrome," "Comparison with Previous Studies," etc.
      • Using bullet points for some statistics or findings can enhance readability.
    • Detailing Methodology:
      • Readers might need more insight into how the diagnoses were made, the criteria used, the selection process of the subjects, etc., to better understand the context of the findings.
    • Comparison with Other Studies:
      • A tabular comparison with other relevant studies may help visually distinguish the differences and similarities.
    • Confusion with Acronyms: The acronym ASD is used for both "Acute Stress Disorder" and "Autism Spectrum Disorder." Although the focus appears to be on Acute Stress Disorder in this context, using the same acronym for both could lead to confusion. Consider using unique acronyms or explicitly stating which disorder is being referred to in each instance.

    • More Detail on Statistical Methods: The discussion could benefit from references to percentages and statistics, but it might benefit from more detailed information about the statistical methods used to arrive at these conclusions. This would add robustness to the claims and facilitate replication by other researchers.
    • Clarification of Terminology: Some terms or concepts, such as the NSESSS scores and the sixth or third item mentioned, might need further explanation or context. Without previous information, readers may find these points unclear.
    • Inclusion of Limitations: Though the section does call for more extensive studies, it might benefit from a more explicit discussion of the limitations of the current research. This could include potential biases, the limited scale of the study, or any other factors that might limit the generalizability of the findings.
    • Discussion of Non-Pharmacological Treatments: While the focus on pharmacological treatment is vital, including a discussion of non-pharmacological interventions, if applicable, could provide a more rounded perspective. This might involve considering therapy, counselling, or community support alongside medication.
    • Enhancing Flow and Structure: Breaking the paragraph into sub-sections with appropriate headings and subheadings could strengthen readability. The content appears to cover several aspects like pharmacological treatments, symptom analysis, parent-child agreement, etc. Structuring these under relevant headings could make the information more digestible for the reader.
    • Patient and Family Perspective: While this may not be the paper’s focus, incorporating some insights into the patients' and families' experiences or perceptions of the treatment might provide a more human-centred view of the subject matter.
    • Clarity and Detail:
      • The section on fathers' education level and its correlation with ASD symptoms in children could benefit from further elaboration. The underlying mechanisms are described but could be more evident with more concise and detailed explanations.
      • Explaining the NSESSS scores and how children without physical injuries might have higher scores is complex and could be more clearly described. More context about what NSESSS scores represent helps in understanding this point.
    • Methodological Limitations:
      • The text already mentions some limitations, such as the restriction to adolescents who sought care at the outpatient clinic and the absence of examining ASD symptoms in the parents. Expanding on these limitations and how they affected the results could add depth to the analysis.
      • Suggestions for how future studies might overcome these limitations strengthen this section.
    • Implications and Recommendations:
      • The findings significantly affect how trauma is addressed in the aftermath of natural disasters. The section could benefit from specific recommendations on how these findings can be applied, such as targeted mental health interventions for children without physical injuries and for children of highly educated fathers.
      • Discussing potential policy implications and how this research might inform emergency response strategies would add value.
    • References and Comparison with Previous Studies:
      • While the section does provide references to previous studies, it could benefit from a more systematic comparison with prior research, highlighting the unique contributions of this study and how it fits into the broader scientific landscape.
    • Longitudinal Study Mention:
      • The mention of a planned longitudinal study is a positive note, but more details on what this future research might entail and why it's necessary help the reader understand its importance.
    • Language and Structure:
      • Some sentences are quite long and complex, which might affect readability. Breaking them into shorter, more concise statements could make the text more accessible.
      • Organising the findings into clear subheadings, such as "Physical Injuries and ASD," "Fathers' Education Level," and "Limitations and Future Research," could enhance the structure and readability of the section.
    • Emphasizing Psychological Well-Being:
      • While the study emphasizes the need to evaluate psychological well-being without physical injuries, more specific strategies or programs could be elaborated to address this need.

  •  

      •  

    The conclusion is relatively brief, and some areas can be improved:

    • Completeness:
      • The last sentence appears to be cut off and incomplete. Completing that thought would clarify the intended message.
    • Specificity:
      • The conclusion could be strengthened by summarizing the key findings more explicitly. This includes the surprising NSESSS scores in adolescents without physical injuries and the correlation between fathers' education levels and ASD symptoms.
    • Recommendations and Implications:
      • Although the need for targeted interventions is mentioned, specific recommendations and practical applications of the findings could be highlighted. For example, outlining how the results can inform mental health policies, emergency response strategies, and healthcare professional training.
    • Limitations:
      • While not always necessary in conclusion, acknowledging the study's limitations and how they might affect the interpretation of the results can enhance the reader's understanding.
    • Connection to Broader Context:
      • The conclusion could relate the study's findings to the broader context of mental health care following natural disasters, emphasizing why the study's insights are significant and timely.
    • Future Research:
      • While there's a mention of the need for further research, outlining specific areas or questions that future research should address could provide a more precise direction. This might include longitudinal studies, different demographic evaluations, or novel therapeutic approaches.
    • Language and Tone:
      • The language used in the conclusion is appropriate but could benefit from a more engaging and compelling tone to emphasize the significance of the findings.
    • Connection to Introduction and Objectives:
      • Reinforcing the study’s primary objectives and how the results met (or did not meet) those objectives can provide a full-circle understanding for the reader.

I recommend the following revisions related to the English language:

  • Grammar and Punctuation: Please revise the manuscript for grammatical accuracy and consistent punctuation usage, particularly around citations.
  • Spelling and Typographical Errors: Thorough proofreading is required to correct typographical errors and ensure proper spelling throughout the manuscript.
  • Clarity and Coherence: Consider revising complex sentences to improve readability and clarity.
  • Consistency in Terminology: Ensure that terms are defined when first introduced and consistent terminology is used.

Round 2

Reviewer 1 Report

The author has further revised the manuscript in response to my concerns. I recommend accepting this version for the current edition.

Reviewer 2 Report

The authors have adequately addressed the comments they could through revisions and have provided proper justification for those they could not or did not agree with my review comments.

Minor editing of English language required